# Approach to the Connection between Meconium Consistency and Adverse Neonatal Outcomes: A Retrospective Clinical Review and Prospective In Vitro Study

**DOI:** 10.3390/children8121082

**Published:** 2021-11-24

**Authors:** Hueng-Chuen Fan, Fung-Wei Chang, Ying-Ru Pan, Szu-I Yu, Kuang-Hsi Chang, Chuan-Mu Chen, Ching-Ann Liu

**Affiliations:** 1Department of Pediatrics, Tungs’ Taichung Metroharbor Hospital, Wuchi, Taichung 435, Taiwan; fanhuengchuen@yahoo.com.tw; 2Department of Medica research, Tungs’ Taichung Metroharbor Hospital, Wuchi, Taichung 435, Taiwan; liz00049@yahoo.com.tw (Y.-R.P.); pride1223@gmail.com (S.-I.Y.); kuanghsichang@gmail.com (K.-H.C.); 3Department of Rehabilitation, Jen-Teh Junior College of Medicine, Nursing and Management, Miaoli 356, Taiwan; 4Department of Life Sciences, Agricultural Biotechnology Center, National Chung Hsing University, Taichung 402, Taiwan; chchen1@dragon.nchu.edu.tw; 5Department of Obstetrics and Gynecology, Tri-Service General Hospital, National Defense Medical Center, Taipei 11490, Taiwan; doc30666@gmail.com; 6The iEGG and Animal Biotechnology Center, and Rong Hsing Research Center for Translational Medicine, National Chung Hsing University, Taichung 402, Taiwan; 7Bioinnovation Center, Buddhist Tzu Chi Medical Foundation, Hualien 970, Taiwan; 8Department of Medical Research, Hualien Tzu Chi Hospital, Hualien 970, Taiwan; 9Neuroscience Center, Hualien Tzu Chi Hospital, Hualien 970, Taiwan

**Keywords:** meconium-stained amniotic fluid (MSAF), meconium aspiration syndrome (MAS), cyclooxygenase-2 (COX-2), nitric oxide (NO), nitric oxide synthase (NOS)

## Abstract

Whether meconium-stained amniotic fluid (MSAF) serves as an indicator of fetal distress is under debate; however, the presence of MSAF concerns both obstetricians and pediatricians because meconium aspiration is a major contributor to neonatal morbidity and mortality, even with appropriate treatment. The present study suggested that thick meconium in infants might be associated with poor outcomes compared with thin meconium based on chart reviews. In addition, cell survival assays following the incubation of various meconium concentrations with monolayers of human epithelial and embryonic lung fibroblast cell lines were consistent with the results obtained from chart reviews. Exposure to meconium resulted in the significant release of nitrite from A549 and HEL299 cells. Medicinal agents, including dexamethasone, L-Nω-nitro-arginine methylester (L-NAME), and NS-398 significantly reduced the meconium-induced release of nitrite. These results support the hypothesis that thick meconium is a risk factor for neonates who require resuscitation, and inflammation appears to serve as the primary mechanism for meconium-associated lung injury. A better understanding of the relationship between nitrite and inflammation could result in the development of promising treatments for meconium aspiration syndrome (MAS).

## 1. Introduction

Meconium is a black-green, odorless, rather sticky, and viscous material that can be found in the bowel of the developing fetus starting from 70–85 days of gestation [1,2,3]. Meconium contains bile acids and salts, mucus, pancreatic juices, cellular components exfoliated from the gastrointestinal tract, swallowed amniotic fluid, vernix caseosa, lanugo hairs, mucus glycoproteins, lipids, proteases, and blood that accumulates in the fetal colon throughout gestation [4,5]. When the meconium becomes excreted into the amniotic cavity, meconium-stained amniotic fluid (MSAF) can be detected [6,7,8,9,10]. MSAF can serve as an indicator of fetal bowel maturation [11] and can also represent a secondary fetal distress sign due to hypoxia [12,13,14]. Animal studies have revealed that hypoxia evokes a vagal response, stimulating colonic activity and relaxing the anal sphincter, promoting the release of meconium into the uterine cavity [15]. Animal studies also showed that fetal swallowing was suppressed by hypoxia, leading to a decrease in the normal ability to clear meconium from the amniotic fluid [16]. Therefore, hypoxia may result in excessive meconium excretion, disturb clearance, and prolong MSAF, which is associated with intrauterine fetal death, low APGAR scores [17], intrapartum fetal death [18], neurologic impairments [19,20], and meconium aspiration syndrome (MAS) [21].

Approximately 1% to 12% of neonates with MSAF will develop MAS [22,23,24], which is associated with various serious complications, such as persistent pulmonary hypertension (PPHN), long-term respiratory issues [7,25,26], neurodevelopmental problems [17,19,20,27,28,29], and mortality [6]. MAS is a multifaceted disease, characterized by airway obstruction, surfactant dysfunction, and pulmonary inflammation [30]. Aspirated meconium that obstructs the airway impacts the infant’s oxygenation capacity [20,21], leading to the development of pneumothorax [22], pulmonary hypertension [23], and chemical pneumonitis [24], all of which can contribute to the occurrence of severe acute hypoxia, impaired neural development, and death [25,26]. However, routine intubation with suction is no longer recommended for the removal of meconium because these interventions have not been demonstrated to significantly reduce the incidence of MAS or MAS-related mortality [31,32], suggesting that other mechanisms may be responsible beyond airway obstruction.

Aspirated meconium can directly damage type II pneumocytes [24,33], and the enzymes found in meconium can cleave surfactants [33], leading to a significant decrease in surfactant levels. Moreover, aspirated meconium can alter surfactant fluidity [34] and ultrastructure [24], resulting in surfactant dysfunction. Although the administration of exogenous surfactant improved lung functions in an animal model of MAS [35], this approach is supported by limited data, and clinical trials of exogenous surfactant administration did not show significant reductions in MAS-associated mortality or other morbidities [36,37]. An important feature of newborn lungs exposed to meconium is the presence of an inflammatory response [38], in which inflammatory cells and cytokines, such as tumor necrosis factor (TNF)-α, interleukin (IL)-1*β*, IL-6, and IL-8, are activated by meconium to initiate pulmonary inflammation [30], and increased inflammatory indices are detected in cases of severe MAS [39]. Pathological examinations in MAS cases have revealed typical inflammatory pneumonitis, characterized by epithelial disruption, proteinaceous exudation with alveolar collapse, and cellular necrosis [40]. Together, these findings, combined with the clinical features of MAS [9,24,41,42,43], suggest that meconium causes profound functional alterations within the lungs, associated with an intense inflammatory reaction [33].

Nitric oxide (NO) is a ubiquitous gas that is involved in diverse physiological processes, including vasodilation, bronchodilation, neurotransmission, tumor surveillance, antimicrobial defense, and the regulation of inflammatory-immune processes [44,45,46]. Although inhaled NO can successfully treat MAS associated PPHN [47], NO inhalation was only associated with transient decreases in airway resistance and pulmonary pressure in animal models of MAS, suggesting that the underlying mechanisms associated with MAS extends beyond abnormal vascular constriction and may involve the lung parenchyma [48]. Moreover, NO can potentiate lung injury by promoting oxidative or nitrosative stress [49], inactivating surfactants, and stimulating inflammation [50]. NO is generated from L-arginine by three different NO synthases (NOS): neuronal NOS (nNOS; NOS-1), inducible NOS (iNOS; NOS-2), and endothelial NOS (eNOS; NOS-3) [51]. The role played by NO in meconium-induced lung injury remains unclear.

A pilot randomized control trial demonstrated a lack of significant differences in the outcomes of mild, moderate, and severe MAS when comparing cases treated with or without endotracheal suction [52], suggesting that meconium consistency has no effect on MAS prognosis; however, based on our own clinical experience, we suspected hypothesized that a potential connection exists between meconium consistency and MSAF prognosis. To investigate this hypothesis, we first examined the clinical data of neonates born with meconium from a local teaching hospital. Furthermore, we developed an in vitro model using human alveolar epithelial and bronchial cells to determine the effects of different meconium concentrations on lung cells.

## 2. Materials and Methods

### 2.1. Human Study 

#### Data Sources

The medical records associated with live births delivered at Tungs’ Taichung MetroHarbor Hospital between 1 January 2013 and 31 December 2017 were reviewed, including the paper and electronic records of all infants admitted to the nursery, the sick neonate care unit, and the neonatal intensive care unit (NICU). Diagnoses were determined by qualified pediatricians according to the International Classification of Diseases, Clinical Modification, 9th Revision (ICD-9CM). Meconium consistency was categorized as either thick (dark green in color and with a pea soup consistency) or thin (lightly-stained yellow or greenish color) [53]. All enrolled subjects were de-identified and encrypted by the manager of the medical record at Tungs’ Taichung MetroHarbor Hospital to protect patient privacy, and these data cannot be used either to trace individual patients or be linked to other census data, such as the cancer registry or the household registry. Due to the anonymized nature of the dataset, the need for informed consent was waived. This study was approved by the institutional review board at Tungs’ Taichung MetroHarbor Hospital, Taiwan, ROC (IRB approval No.: 107048). All protocols used in the human study were performed in accordance with the ethical standards established by the 1964 Declaration of Helsinki and its later amendments or comparable ethical standards [54].

### 2.2. Cell Study

#### 2.2.1. Preparation of Meconium

As the birth canal is not a sterile environment [55,56,57,58,59], we collected meconium from ten full-term, healthy neonates delivered via cesarean section to minimize potential contamination during delivery. Meconium was prepared according to a previously published method [60]. In brief, we obtained first-pass meconium samples within 30 min of passage, which were transferred from the diaper into a sterile container. These samples were pooled together and processed in a blender to achieve a uniform consistency. After being homogenized with 0.9% NaCl to a 20% (*w*/*v*) final concentration, the meconium was centrifuged at 5000 RPM for 20 min at 4 °C, the supernatant was filtered through an 8-µm filter (Millipore Co., Bedford, MA, USA), aliquoted into 2-mL sterile plastic bottles, and stored at –80 °C until use. For meconium collection, a parent’s or guardian’s permission and informed consent were required. This study was approved by the institutional review board at Tungs’ Taichung MetroHarbor Hospital, Taiwan, ROC (IRB approval No.: 105047). All protocols used during the meconium collection process were performed in accordance with relevant guidelines and regulations [61].

#### 2.2.2. Culture of Lung Cells

Alveolar epithelial cells from the human lung carcinoma cell line A 549 and lung cells from the human embryonic bronchial fibroblast cell line HEL 299 were purchased from the American Type Culture Collection (Manassas, VA, USA). All cells tested negative for *Mycoplasma* contamination before any experiments were conducted in this study. These cells were grown in monolayers at 37 °C in 5% CO_2_ and 100% humidity using tissue culture dishes. A549 cells were maintained on RPMI1640 (Gibco BRL, Grand Island, NY, USA). HEL299 cells were maintained on Modified Eagle’s Medium (MEM; Gibco BRL, Grand Island, NY, USA). Both media were supplemented with penicillin (1 × 10^5^ U/L), streptomycin (100 mg/L), amphotericin B (0.25 mg/L), 2 mM L-glutamine (Invitrogen, Carlsbad, CA, USA), and 10% (*v/v*) fetal bovine serum (FBS, Hyclone Laboratories, Logan, UT, USA). The same batch of FBS was used for all experiments. The culture medium was renewed every 2–3 days.

#### 2.2.3. Meconium Stimulation

A549 and HEL 299 cells were plated into 96-well culture plates at a concentration of 1 × 10^5^ cells/mL and incubated at 37 °C in 5% CO_2_ for 24 h. After washing, A549 and HEL299 cells were incubated for an additional 24 h with serum-free RPMI1640 and MEM, respectively. A preliminary study showed that the percentages of cell death were similar when cells were exposed to meconium concentrations ≥20% at different time points (data not shown). Therefore, 20% meconium was used as a stock solution and was diluted with RPMI1640 or MEM to obtain various concentrations (0.1%, 1%, and 5%). Monolayers of cells were then incubated in a meconium-containing medium for various periods of time (1, 6, 12, 18, and 24 h). Control cells were incubated in a meconium-free medium in a similar manner. At each time point, the supernatant was collected and used to determine cell viability and nitrite production. The cells were washed twice with phosphate-buffered saline (PBS) and collected for RNA extraction. 

#### 2.2.4. Cell Viability

Cell viability was analyzed by measuring the activity of mitochondrial malate dehydrogenase (mMDH) using the WST-1 assay [62]. A549 and HEL299 cells were plated in 96-well plates, treated with or without meconium stimulation, and incubated with 10 μLof WST-1 reagent (BioVision, Milpitas, CA, USA) for 3 h at 37 °C. The amount of formazan generated, which was proportional to the number of viable cells, was calculated using a Multiskan™ FC Microplate Photometer (Molecular Devices) based on the absorbance signal at 440 nm. The absorbance was corrected using a background reading.

#### 2.2.5. Nitrite Determination

Nitrite production was measured by a Griess assay, as previously described [63]. Briefly, the concentration of nitrite in A549 and HEL299 cells treated with or without meconium stimulation in the absence or presence of 2 mM L-NAME; 10^−4^, 10^−6^, 10^−8^, or 10^−10^M dexamethasone; or 25, 50, or 100 µM NS-398 in each well were measured by adding 100 μL Griess reagent (0.1% N-(1-Naphthyl) ethylenediamine in dH2O and 1% sulfanilamide in 5% (*v*/*v*) phosphoric acid, mixed 1:1 immediately before use) to 100 μL of culture supernatant, followed by incubation at room temperature for 10 min. The absorbance at 540 nm was measured using a Multiskan™ FC Microplate Photometer (Molecular Devices). Nitrite concentrations in the culture supernatant were calculated based on a standard curve using known concentrations of sodium nitrite. The absorbance values were corrected using a background reading.

#### 2.2.6. RNA Extraction and Real-Time Quantitative PCR

Total RNA was extracted from cultured A549 and HEL299 cells, isolated, and purified using TRIzol^®^ RNA Isolation Reagents (Invitrogen, Liverpool, NY, USA). For the synthesis of the first-strand cDNA, 2 μg of total RNA was collected for a single-round reverse transcription reaction, performed using a High-Capacity cDNA Reverse Transcription Kit (Applied Biosystems, Foster City, CA, USA). cDNAs were exponentially doubled under conditions of 95 °C for 30 s, 40 cycles at 95 °C for 1s, and 60 °C for 60 s, using the TaqMan probes PCR master mix (Applied Biosystems) and a Step-One™ Real-Time PCR System (Applied Biosystems). The simultaneous amplification of β_2_-microglobulin (B2M) was used as an internal control against which to normalize the various mRNA levels in the samples and to quantify changes in gene expression levels using the 2^−ΔΔCt^ formula. The specific primers used in this study are shown in Table 1. All reactions were performed in at least triplicate and normalized to B2M gene expression levels. The data were analyzed using Bio-Rad CFX Manager 3.1 software (Bio-Rad) and are presented as fold changes in the normalized mRNA amounts of the meconium treatment group relative to those of the control group.

#### 2.2.7. Library Preparation and Sequencing 

The purified RNA was used to prepare a sequencing library using the TruSeq Stranded mRNA Library Prep Kit (Illumina, San Diego, CA, USA), following the manufacturer’s recommendations. Briefly, mRNA was purified from total RNA (1 µg) by oligo (dT)-coupled magnetic beads and fragmented into small pieces under an elevated temperature. The first-strand cDNA was synthesized using reverse transcriptase and random primers. After the generation of double-strand cDNA and the adenylation of the 3′ ends of DNA fragments, the adaptors were ligated and purified using the AMPure XP system (Beckman Coulter, Beverly, Brea, CA, USA). The quality of the libraries was assessed using the Agilent Bioanalyzer 2100 system and a real-time PCR system. The qualified libraries were then sequenced on an Illumina NovaSeq 6000 platform with 150 bp paired-end reads, generated by Genomics, BioSci & Tech Co., New Taipei City, Taiwan.

#### 2.2.8. Bioinformatics

Low-quality bases and sequences from adapters were removed from the raw data using the program Trimmomatic (version 0.39). The filtered reads were aligned to the reference genomes using Bowtie 2 (version 2.3.4.1). A user-friendly software, RSEM (version 1.2.28), was applied for the quantification of transcript abundance. Differentially expressed genes (DEGs) were identified by EBSeq (version 1.16.0) [64].

#### 2.2.9. Statistical Analysis

Summary statistics are expressed as the frequency and percentage for categorical data and as the mean and standard deviation (SD) for continuous variables. Group differences in the distribution of delivery mode, preeclampsia, diabetes, antepartum hemorrhage, PROM, polyhydramnios, oligohydramnios, sex of the infant, hypoglycemia, NICU admission, CRAP use, intubation, ventilator use, and death were analyzed by the Fisher’s exact test. Continuous variables, such as APGAR scores and maternal and gestational age, were compared between the thin and thick meconium groups using the Student’s *t*-test. The survival percentages and the effects in cells exposed to various concentrations of meconium (0.5%, 1%, and 5%) and various treatment durations (1, 6, 12, 18, and 24 h) were evaluated using a one-way analysis of variance (ANOVA). The mRNA expression levels in cells with and without meconium treatment were analyzed by the paired Student′s *t*-test. The nitrite levels in cells exposed to various meconium concentrations (0.5%, 1%, and 5%) for various treatment durations (1, 6, 12, 18, and 24 h), combined with various medicinal agents, were evaluated by one-way analysis of variance (ANOVA). A *p*-value < 0.05 was considered significant for all analyses (* *p*  <  0.05 and ** *p*  <  0.005). Statistical analyses were conducted using the statistical package SAS version 9.4 (SAS Institute Inc., Cary, NC, USA).

## 3. Results

### 3.1. Thick Meconium Is a Risk Factor for Neonates Receiving Resuscitation

A total of 8316 neonates were delivered at a local teaching hospital during this five-year study, including 3078 (37.01%) neonates delivered by cesarean section and 5238 (62.99%) neonates delivered vaginally. The charts for 1099 (13.22%) neonates recorded MSAF, or meconium-stained skin, nail, or umbilicus, including 454 (41.31%) neonates delivered by cesarean section and 645 (58.69%) neonates delivered vaginally. Among these, 95 (1.14%) neonates were deemed to have suffered from MAS and were admitted to the sick neonate care unit, and 12 neonates were admitted to the NICU. The male:female ratios were 598:501 in the MSAF group and approximately 1:1 (48: 49) in the MAS group. 

To investigate the effects of exposure to different meconium consistencies among infants diagnosed with MAS, we divided the infants diagnosed with MAS into thin and thick meconium groups, based on the data obtained from the chart review, resulting in 72 cases classified into the thin meconium group and 23 cases classified into the thick meconium group. No significant differences were identified among maternal factors such as maternal age; delivery mode; or medical conditions, such as preeclampsia, diabetes, antepartum hemorrhage, PROM, polyhydramnios, and oligohydramnios. Several neonatal factors, including gestational age, birth, weight, sex of the infant, and the prevalence of hypoglycemia did not differ significantly between the two groups. However, the APGAR scores at 1 min and 5 min, the numbers of neonates who required NICU admission, CPAP use, intubation, or ventilator use, and the number of neonates who died showed significant differences between the two groups, suggesting that the presence of thick meconium may be significantly associated with receiving advanced life support (Table 2).

### 3.2. Thick Meconium with Longer Exposure Times Induces Lung Cell Death

The results of the cell viability assay demonstrated that A549 (Figure 1A) and HEL299 cells (Figure 1B) showed different responses following exposure to variable meconium concentrations and meconium exposure durations. Furthermore, we found that higher meconium concentrations or longer exposure times resulted in increased cell death, suggesting that the concentration and exposure time had a significant effect on lung cell viability.

### 3.3. Meconium Induces NOS and COX Gene Expression

A significant amount of cell death was observed when A549 (Figure 1A) or HEL299 cells (Figure 1B) were exposed to meconium at concentrations higher than 0.1% with an exposure time equal to or longer than 6 h. To investigate the effects of meconium exposure on lung cell gene expression, a pilot RNA-seq study was performed using RNA samples from A549 and HEL299 cells following a 6 h exposure to 1% meconium. Meconium may activate inflammatory cells and induce cytokines to initiate pulmonary inflammation [24], and NOS and COX, which are the primary inflammatory mediators and are expressed in the airway epithelium, releasing NO and COX products during acute inflammatory responses [65]. The results of the RNA-seq showed greater fold changes in nitrite production-related genes, including *NOS-1*, and *NOS-2*, especially *NOS-2*, in the HEL299 cells than in A549 cells. The *COX-2* expression levels in both A549 and HEL299 cells were very high (Table 3). These genes were selected for further study, and their expression was validated by real-time RT-PCR. *NOS-1*, *NOS-2*, *NOS-3*, and *COX-2* expression was detectable in both A549 and HEL299 cells. Significant differences in *NOS-1* mRNA expression levels were observed between the HEL299 cells with meconium stimulation versus those without (Figure 2A). However, no significant differences in *NOS-1* mRNA expression levels were observed in A549 cells with or without meconium stimulation (Figure 2A). *NOS-2* (Figure 2B) and *COX-2* (Figure 2D) mRNA levels increased significantly following meconium stimulation in both in A549 cells and HEL299 cells. No significant differences were observed for *NOS-3* mRNA levels in A549 and HEL299 cells with and without meconium stimulation (Figure 2C).

### 3.4. Meconium Enhances Nitrite Production

Nitrite levels from A549 (Figure 3A) and HEL299 cells (Figure 3B) exposed to meconium at concentrations higher than 0.1% increased significantly compared with those in the control cells. Using 1% meconium, nitrite production significantly increased in the supernatant collected from A549 and HEL299 cells after 1, 6, 12, 18, and 24 h exposure compared with nitrite levels in the control cells. The results also showed that the nitrite production by HEL299 cells was significantly greater than that observed for A549 cells after 6 h of exposure to 1% meconium (nitrite levels in HEL299 after 6 h vs. nitrite levels in A549 after 6 h: 411.18 ± 36.41 vs. 238.13 ± 22.29, *p* = 0.017). 

### 3.5. Dexamethasone and COX-2 Inhibitor Treatment Significantly Reduced the Nitrite Production Induced by Meconium Stimulation

The effects of various medicinal agents on nitrite production were examined. The addition of 2 mM arginine increased nitrite production in HEL299 cells to a greater degree than in A549 cells following meconium exposure. The nitrite levels observed in HEL299 and A549 cells treated with 2 mML-NAME; 10^−10^, 10^−8^, 10^−6^, or 10^−4^M dexamethasone; or 25, 50, or 100 µM NS-398. L-NAME, dexamethasone, and NS-398 treatment were all able to significantly reduce nitrite production in A549 (Figure 4A) and HEL299 cells treated with 1% meconium for 6 h (Figure 4B).

## 4. Discussion

The presence of MSAF raises serious concerns, and meconium aspiration remains a major contributor to neonatal morbidity and mortality, despite appropriate treatment strategies [7,66]. In this study, the incidence of MSAF was 13.22%, which is within the reported range from 5–20% [6,7,8,9,10]. MAS is diagnosed in neonates born through MSAF who present with symptoms that cannot be otherwise explained [67]. In this study, the incidence rate of MAS was 1.14% among all neonates, which was within the reported range from 0.2–1.3% in China [68,69]. In the USA, the incidence rates for MAS range from 0.1 to 0.4% of births [29,70]. In France, the incidence of MAS was reported to be 0.2% [22]. In this study, 8.64% of neonates with MSAF exhibited airway symptoms, and 6.31% of MAS diagnosed neonates required ventilator support, in addition to an MAS diagnosed mortality rate of 2.1%. The literature has reported that between 4.2 and 62% of infants born through MSAF subsequently suffer from respiratory distress [22,71], and between 33 and 49.7% of MAS diagnosed neonates require ventilator support, with a 5–12% mortality rate [52,72,73]. The discrepancy between our results and those reported by others may be due to differences in ethnicity, socio-demographic variables, health institutions, or the provided level of medical care reported across different countries, case numbers, and study time points.

Although numerous studies have reported no significant differences in adverse neonatal outcomes associated with meconium consistency [52,74,75,76], our clinical experience [77], and the results of this retrospective analysis suggest that thick meconium may serve as a clinical risk for adverse neonatal outcomes, including low APGAR scores at 1 min and 5 min, neonatal death, or the need for NICU admission, resuscitation, CPAP use, or ventilator use. Our findings agree with the results of several papers [4,21,22,78,79,80]. Maternal factors, including maternal age, delivery mode, and the presence of medical conditions such as preeclampsia, diabetes, antepartum hemorrhage, PROM, polyhydramnios and oligohydramnios, and neonatal factors including gestational age, birth weight, sex of the infant, and the prevalence of hypoglycemia have all been reported to be associated with the presence of MSAF [14,21,22,25,78,81,82,83,84,85]. Because thin meconium is reported to be associated with chronic hypoxic stress, whereas thick meconium is reported to be associated with acute hypoxic stress or inflammation [13,79], we hypothesize that fetal asphyxia that occurs before or during delivery in both groups may represent a confounding factor that might neutralize the power of statistical analyses in both groups, leading to the lack of significant differences in maternal factors between the two groups. 

Clinically, our results and those reported by others [14,21,22,25,78,81,82,83,84,85], showed that the presence of thick meconium was associated with higher rates of respiratory compromise, intubation at birth, and receiving ventilator support compared with thin MSAF. As a result, infants with thick MSAF have higher exposures to acute hypoxic events, leading to a higher risk of developing respiratory insufficiency. However, an interesting study proposed that the extent of lung destruction observed in MAS was not related to the aspiration of meconium but rather to the length and degree of asphyxia [86]. Infants who experience long and severe asphyxia usually demonstrate airway symptoms soon after delivery; however, some cases of MAS in this study and another study [87] developed in apparently healthy, meconium-stained neonates. Although we are not able to determine the occurrence of potential fetal asphyxia before or during labor among infants with MSAF, we hypothesize that inflammation in the airway may occur when an MSAF infant develops MAS, especially in the thick meconium group. The processes underlying inflammation in the airway at the cellular level are less well understood. 

Our study showed that the cellular viabilities of both alveolar epithelial and bronchial cells were significantly reduced by stimulation with human meconium, and the severity of the response correlated with both exposure time and the meconium concentration, suggesting that meconium exerts a direct toxicity effect in alveolar and bronchial cells. Although bile salts and proteolytic enzymes, which are considered toxic components found in meconium, are capable of injuring the alveolar and bronchial structures [4,5], the consequent inflammation triggered by the meconium in the lungs may explain “postsurfactant slump” [88].

In this study, the results of the Figure 1, Figure 3, and Figure 4 demonstrated that A549 cells and HEL299 cells presented with different responses to the same meconium stimuli, suggesting that alveolar and bronchial cells might respond differently to meconium exposure, which was also compatible with the results of our RNA-seq and RT-PCR results. Although several papers have used A549 cells treated with meconium to simulate MAS in vitro [6,42,60,89], and A549 cells retain some characteristics of normal alveolar type II cells, A549 cells are in fact an adult human lung carcinoma cell line [90]. Additionally, bronchial tissue has been reported to be involved in the pathogenesis of MAS [50,91]; therefore, in this study, HEL299 cells, which are derived from fetal bronchial tissue, were used in combination with A549 cells to explore the effects of exposure to various concentrations of meconium.

RT-PCR showed that the mRNA expression of *NOS-1* was slightly elevated in A549 cells following 1% meconium exposure for 6 h, but this effect was not significant. Although NOS-3 is viewed as an important regulator of nitrite production in the perinatal lung vasculature [92], A549 and HEL299 cells do not contain any vascular components, so the changes in the mRNA expression levels of *NOS-3* were not significant following meconium stimulation. The levels of *NOS-2* mRNA significantly increased following meconium stimulation in both A549 cells and HEL299 cells. We hypothesize that *NOS-2* serves as the primary nitrite production mechanism in A549 cells in response to meconium stimulation, which is supported by the literature [93,94]. Moreover, in this study, the mRNA levels of *NOS-1* and *NOS-2* were significantly higher in HEL299 cells than in A549 cells in response to meconium stimulation, suggesting that fetal HEL299 cells could generate more nitrite than A549 cells even under identical meconium exposure conditions. 

Nitrite production significantly increased in the supernatant derived from both A549 and HEL299 cells after 1, 6, 12, 18, and 24 h of 1% meconium exposure compared with nitrite production by control cells. Consistently, the amount of nitrite produced by HEL299 cells treated with 1% meconium for 6 h was significantly greater than that observed for A549 cells (Figure 3**)**. Therefore, the association between elevated nitrite levels and reduced lung cell viability following meconium stimulation suggests that NO may play an important role in the pathogenesis of meconium-associated lung injury. The involvement of NO in meconium-associated lung injury has previously been studied [41,95,96,97,98,99]. Although the idea that elevated levels of NO might contribute to tissue damage is not new, our results showed HEL299 cells generated higher nitrite levels than A549 cells (Figure 3); therefore, previous experiments using A549 cells may not accurately represent the extent of lung injuries caused by meconium exposure. 

Various studies of animals and cell lines examining the effects of meconium exposure have indicated the involvement of inflammatory mediators, such as NO and COX [41,95,96,97,98]. NO is required for vasodilation in PPHN and may cause other pathological changes in the body associated with the activation of inflammatory cells and cytokines, especially during lung injury [100,101]. In addition to NO, COX, also known as prostaglandin synthase, is a potent inflammatory mediator. Two mammalian COX enzyme isoforms have been identified, COX1 and COX2, which are considered constitutive and inducible, respectively [102]. The anti-inflammatory effects of non-steroidal anti-inflammatory drugs primarily act through their abilities to inhibit prostaglandin production, particularly through the inhibition of COX-2 activity [103]. COX is similar to NOS, including the expression of both constitutive forms, which are mostly involved in housekeeping tasks [104], and inducible forms, which shape the cellular response to stress and various bioactive agents [105]. For example, both NOS and COX in the airway epithelium become activated during acute inflammatory responses [65]. A number of studies have also suggested a role for COX in the cytotoxic effects of MAS [97,106]. Our RNA-seq and RT-PCR data showed that COX-2 mRNA levels in both A549 and HEL299 cells were highly expressed in response to meconium stimulation, and NS-398, a COX-2 specific inhibitor, has been shown to inhibit inflammation-related COX-2 activity [107]. Our results showed that the anti-inflammatory effect of NS-398 mitigated meconium-induced COX-2 over-expression, which, in turn, reduced the meconium-induced nitrite production in both A549 and HEL299 cells, suggesting that NS-398 may have an inhibitory effect against the cytotoxic effects of MAS. Furthermore, these data suggest that inflammation may represent a primary mechanism underlying lung injury induced by meconium aspiration.

L-NAME is a competitive inhibitor of NOS [108] and was able to prevent the release of NO from A549 and HEL299 cells in response to meconium stimulation. Although L-NAME has been shown to significantly decrease the levels of both nitrite and nitrate in cellular supernatants [96,109], L-NAME is also associated with teratogenic fetal limb defects and cannot be used for the treatment of MAS. As lung cells may produce NO from L-arginine due to NOS activity, L-arginine in this study was used as a positive control. Although corticosteroids are among the most effective anti-inflammatory agents used to treat many inflammatory diseases [110], the use of steroids is not recommended by the Cochrane database for the treatment of MAS [111], and a meta-analysis showed that steroid use did not decrease mortality or associated morbidities [112]. However, our experience and those reported by others have indicated that the outcomes of infants with MAS can be significantly improved by the administration of both systemic and inhaled steroids [113,114,115]. Intratracheally instilled steroid, either alone or with a surfactant, has resulted in a good response in an animal MAS model [116,117], and the use of steroids significantly attenuated pulmonary hemodynamic deterioration and structural lung damage caused by meconium aspiration in a piglet MAS model [50]. Moreover, NO and COX-2 production were inhibited by steroid treatment [118,119]. In this study, the nitrite levels induced by meconium exposure in both cell types could be significantly reduced by dexamethasone treatment, at a concentration as low as 10^−10^ M, suggesting that the use of dexamethasone may potentially protect against MAS-induced inflammation.

Because inhaled NO(INO) has a potent vasodilating effect, INO was approved by the FDA in 2000 as an effective regimen for the treatment of infants with MAS-associated PPHN [47,92]. Additionally, INO displays anti-inflammatory effects, including reducing cytokine synthesis, inactivating nuclear factor -κB (NF-κB), decreasing the expression of adhesion molecules, and preventing neutrophil adhesion and migration to the alveolar space [120]. In this study, NO, nitrite, nitrate, and NO-derived metabolites were generated when lung cells were stimulated with meconium. Although the generation of NO may be beneficial to lung cells, in plasma or other physiological fluids or buffers, NO becomes almost completely oxidized into nitrite, which remains stable for several hours [121]. Therefore, in many cases, the NO status in the blood does not accurately reflect the corresponding NO status of tissues of interest due to the use of different analysis tools and different samples [122]. Our results suggested that the mechanism underlying the meconium stimulation of lung cells involves inflammation, and meconium stimulation causes a significant decrease in lung cell proliferation (Figure 1). We suspect that the NO generated in this study may only represent a small portion of the generated nitrite found in lung cells stimulated by meconium. Therefore, the effects of INO on lung cells likely differ from the effects reflected by the nitrite data in this study. 

The current study had a number of limitations. First, this study was performed as a retrospective study, and errors may be reflected in the medical records. Second, the sample size was small, and the duration of follow-up was only five years. Significant reductions in morbidity and mortality may have increased with a longer period of follow-up. Third, this was performed as a single-center study. Multi-center, international studies may provide a more convincing result. Fourth, infants who underwent rescue procedures may have had significantly lower APGAR scores than infants who did not. Therefore, we suggest that the relationship between the thick MSAF group and significantly low APGAR scores may require further investigation using a prospective study that includes more infants with thick MSAF to clarify this issue. Fifth, fetal alveolar cells may represent a better study material for MAS than fetal bronchial cells and adult alveolar lung cells. Sixth, inconsistencies in the mRNA expression levels for *NOS2* in A549 cells between the RNA-seq results (Table 3) and the real-time RT-PCR results (Figure 2B) may be due to an up-regulation in the cellular *NOS2* mRNA levels in response to stimulation, associated with an increase in the number of passages after thawing [123]. Seventh, the lack of an animal study was a limitation of our study, which may have provided a more comprehensive understanding of the effects of meconium on lung injury. Finally, this study was not randomized. These limitations may have introduced some bias during the analysis of the effects of meconium on neonatal lungs.

## 5. Conclusions

The clinical features of MAS are characterized by profound functional alterations within the lung, associated with an intense inflammatory reaction, and thick meconium causes a more severe fetal inflammatory response than thin meconium. Our clinical data showed that undesired morbidities, such as intensive birth resuscitation, ICU admission, intubation, ventilation, and death, which were observed for the thick meconium group, did not appear in the thin meconium group. Our in vitro studies showed that the thick meconium with longer exposure times markedly induced lung cell death and exposure to meconium resulted in the significant release of nitrite from lung cells. Taken together, these study results further confirm the inflammatory effects of meconium on lung cells while also suggesting future avenues of research regarding potential agents for counteracting these effects in infants.

## Figures and Tables

**Figure 1 children-08-01082-f001:**
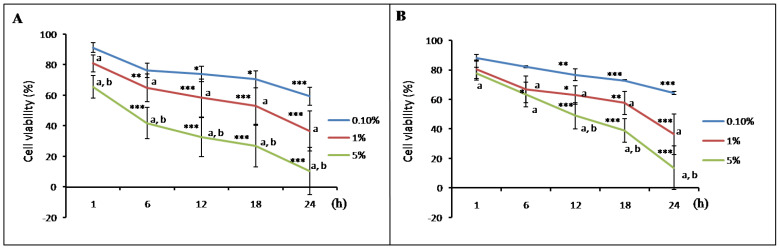
Cell viability of human lung cells, including (**A**) A549: alveolar epithelial cells and (**B**) HEL299: human embryonic lung tissue cells, exposed to 0.1%, 1%, and 5% human meconium concentrations for 1, 6, 12, 18, and 24 h (n_Exp_ = 5). The data represent the mean ± standard deviation. a: *p* < 0.05, versus the 0.1% group; b: *p* < 0.05 versus the 1% group; *: *p*
**<** 0.05 versus at 1 h; **: *p* < 0.005 versus 1 h; ***: *p* < 0.0005 versus 1 h.

**Figure 2 children-08-01082-f002:**
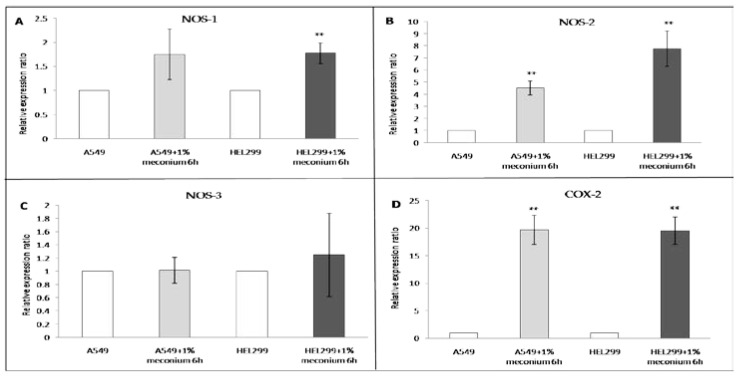
Meconium induces *NOS* and *COX* gene expression. A549 and HEL299 cells were incubated with or without 1% meconium for 6 h. The mRNA expression levels of *B2M* were used as an internal control. Mean relative expression levels for the genes (**A**) *NOS-1*, (**B**) *NOS-2*, (**C**) *NOS-3*, and (**D**) *COX-2*, before and after 1% meconium stimulation for 6 h in A549 and HEL299 cells (n_Exp_ = 4). The data represent the mean ± standard deviations. NOS: Nitric oxide synthases. COX: Cyclooxygenase. **: *p* < 0.005 versus cells without meconium stimulation.

**Figure 3 children-08-01082-f003:**
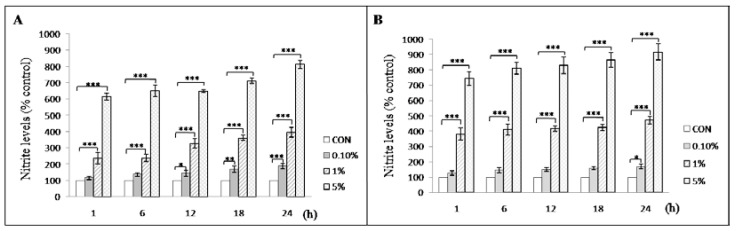
Effects of meconium concentrations on nitrite production in (**A**) A549 and (**B**) HEL299 cells. A549 and HEL299 cells were exposed for 1, 6, 12, 18, or 24 h to 0.1%, 1%, or 5% human meconium. (n_Exp_ = 4). The data represent the mean ± standard deviation. Nitrite levels were significantly higher in the supernatants of cells exposed to meconium compared with the values in control cells. *: *p* < 0.05; **: *p* < 0.005; ***: *p* < 0.0005.

**Figure 4 children-08-01082-f004:**
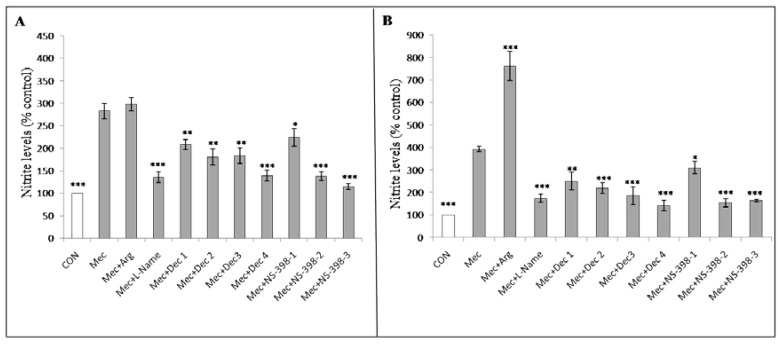
L-NAME, dexamethasone and NS-398 significantly reduced nitrite production following 1% meconium exposure for 6 h in (**A**) A549 and (**B**) HEL299 cells (n_Exp_ = 4). Nitrite levels were significantly higher in the supernatant of untreated cells exposed to meconium compared with those in control cells. The data represent the mean ± standard deviation. Mec: 1% meconium exposure for 6 h. Arg: 2 mM L-arginine; L-NAME: 2 mM L-Nω-nitro-arginine methylester; Dec 1: 10^−10^ M dexamethasone; Dec 2: 10^−8^ M dexamethasone; Dec 3: 10^−6^ M dexamethasone; Dec 4: 10^−4^ M dexamethasone; NS-398-1: 25 µM NS-398; NS-398-2: 50 µM NS-398; NS-398-3: 100 µM NS-398. *: *p* < 0.05; **:*p* < 0.005; ***: *p* < 0.0005.

**Table 1 children-08-01082-t001:** Oligonucleotides primers for real-time RT-PCR analysis.

Gene Name	Sequence	Product (bp)	RefSeq No.
COX-2	Probe 56FAM 5′-ACATCCAGA-ZEN-TCACATTTGATTGACAGTCCA-3IABkFQ-3’	30	NM_000963
	5′- GCCATAGTCAGCATTGTAAGTTG -3′		
	5′- GCACTACATACTTACCCACTTCA -3′		
NOS-1	Probe 56FAM 5′-TCCTTAGCC-ZEN-GTCAAAACCTCCAGAG-3IABkFQ-32032	25	NM_000963
	5′- AGACGCACGAAGATAGTTGAC-3′		
	5′- CCGAAGCTCCAGAACTCAC-3′		
NOS-2	Probe 56FAM 5′- TATTCAGCT -ZEN- GTGCCTTCAACCCCA -3IABkFQ-3′	24	NM_000625
	5′- GCAGCTCAGCCTGTACT-3′		
	5′- CACCATCCTCTTTGCGACA-3′		
NOS-3	Probe 56FAM 5′- TATTCAGCT -ZEN- GTGCCTTCAACCCCA -3IABk FQ-3′	23	NM_001160110
	5′-ACGATGGTGACTTTGGCTA-3′		
	5′-TGGAGGATGTGGCTGTCT-3′		
B2M	Probe 56FAM 5′- CCTGCCGTG -ZEN- TGAACCATGTGACT -3IABkFQ -3′	23	99832111
	5′- ACCTCCATGATGCTGCTTAC -3′		
	5′- GGACTGGTCTTTCTATCTCTTGT -3′		

COX-2: cyclooxygenase-2; NOS-1: nitric oxide synthase-1; NOS-2: nitric oxide synthase-2; NOS-3: nitric oxide synthase-3; B2M: β_2_-microglobulin; RT-PCR: reverse transcriptase-polymerase chain reaction.

**Table 2 children-08-01082-t002:** Comparisons of variables between neonates delivered in the presence of thin or thick meconium.

	Thin Meconium N = 72	Thick Meconium N = 23	*p*-Value
Maternal factors			
Maternal age, years	30.38 ± 4.39	31.13 ± 5.41	0.46
Delivery mode, (CS/NSD)	17/55	8/15	0.29
Preeclampsia, N (%)	5 (6.94%)	2 (8.70%)	0.68
Diabetes, N (%)	6 (8.33%)	2 (8.70%)	1.00
Antepartum hemorrhage, N (%)	2 (2.78%)	1 (4.35%)	0.57
PROM, N (%)	8 (11.11%)	3 (13.04%)	0.72
Polyhydramnios, N (%)	4 (5.56%)	2 (8.70%)	0.35
Oligohydramnios, N (%)	3 (4.17%)	1 (4.35%)	1.00
Neonatal factors			
Gestational age, weeks	39.39 ± 3.01	39.13 ± 2.82	0.51
Birth weight, g	3039.24 ± 497.19	2836.35 ± 490.40	0.09
Sex (female/male)	36/36	12/11	1.00
APGAR1 min	7.80 ± 1.31	6.19 ± 2.64	0.01 *
APGAR5 min	9.01 ± 0.83	7.86 ± 2.22	0.02 *
Hypoglycemia, N (%)	3 (4.17%)	2 (8.70%)	1.00
NICU admission, N (%)	0	12 (52.17%)	<0.001 **
CPAP, N (%)	0	6 (26.09%)	<0.001 **
Intubation, N (%)	0	7 (30.43%)	<0.001 **
Ventilator, N (%)	0	6 (26.09%)	<0.001 **
Death, N (%)	0	2 (8.70%)	0.06

CS: *cesarean* section; NSD: normal spontaneous delivery; PROM: premature rupture of membranes; NICU: neonatal intensive care unit; CPAP: continuous positive airway pressure. *: *p* < 0.05; **: *p* < 0.005.

**Table 3 children-08-01082-t003:** Human lung cell lines were treated with vehicle (1% NaCl) or 1% human meconium for 6 h, and gene expression levels were measured using RNA-seq analysis. Gene expression was analyzed using bioinformatics software to compare expression between meconium-treated and vehicle-treated A549 and HEL299 cells.

Cell Lines		A549	HEL299	
Name/Gene ID/MIM	Gene Description	Fold Increase		Map
NOS1/4842/163731	nitric oxide synthase 1(NOS-1)	0.9845475	1.6171549	12q24.22
NOS2/4843/163730	nitric oxide synthase 2 (NOS-2)	0.4949685	3.2921734	17q11.2
NOS3/4846/163729	nitric oxide synthase 3 (NOS-3)	1.1372183	1.1059593	7q36.1
PTGS2/5743/600262	cyclooxygenase-2 (COX-2)	22.952443	19.439566	1q31.1

## Data Availability

The data are not publicly available due to privacy.

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
