# Peer review of "Approach to the Connection between Meconium Consistency and Adverse Neonatal Outcomes: A Retrospective Clinical Review and Prospective In Vitro Study"

_children, 2021, doi:10.3390/children8121082_

Round 1

Reviewer 1 Report

I read this paper great interest. Authors have studied the effect of thick vs thin meconium on neonatal adverse outcomes. They concluded that thick meconium and prolonged exposure to meconium is associated with need for resuscitation, NICU admission, need for mechanical ventilation and mortality. The postulated mechanism was that thick meconium causes severe inflammation that leads to lung injury.

Major concern: despite interesting topic to read, the In Vitro part of the study (the animal lab/ cell examination) is very hard to follow. I liked the clinical/ practical part of the paper that will help clinician to understand the pathophysiology of MSAF on infant’s lung and airways. I suggest that authors focus their paper on clinical review of MSAF and outcomes. The In Vitro part either summarized in the same paper or referred to it by e-supplement/ or appendix

Other comments

Introduction; very redundant (need to reduced and focused), remove the result of the In Vitro study from the introduction.

Methods.

Data sources: Authors don’t have to mention every single disease code and it is enough to say, “All diagnoses were determined by 128 qualified pediatricians according to the International Classification of Diseases, Clinical 129 Modification, 9th Revision (ICD-9CM).”

Clinical variables: Authors can refer to it by saying as shown in Table 2 (as this unnecessary reptation)

Authors need to add reference/s to indicate what guidelines or regulations they followed “All protocols used in the human study were performed in accordance with relevant guidelines and regulations

Again the in Vitro part of the study need to be separated from the clinical/ retrospective review of data of infants with MSAF to get the message (conclusion) of the study quickly to reader.

Author Response

Dear reviewer 1

Q1: I read this paper great interest. Authors have studied the effect of thick vs thin meconium on neonatal adverse outcomes. They concluded that thick meconium and prolonged exposure to meconium is associated with need for resuscitation, NICU admission, need for mechanical ventilation and mortality. The postulated mechanism was that thick meconium causes severe inflammation that leads to lung injury.

R: Thank you for your very instructive comments.

Q2: Major concern: despite interesting topic to read, the In Vitro part of the study (the animal lab/ cell examination) is very hard to follow. I liked the clinical/ practical part of the paper that will help clinician to understand the pathophysiology of MSAF on infant’s lung and airways. I suggest that authors focus their paper on clinical review of MSAF and outcomes. The In Vitro part either summarized in the same paper or referred to it by e-supplement/ or appendix

R: Thank you for your very instructive comments. We have followed your instructions to revise our manuscript.

Q3: Introduction; very redundant (need to reduced and focused), remove the result of the In Vitro study from the introduction.

R:  Thank you for your very instructive comments. We have followed your instructions to revise our manuscript.

Q4: Methods.

Data sources: Authors don’t have to mention every single disease code and it is enough to say, “All diagnoses were determined by 128 qualified pediatricians according to the International Classification of Diseases, Clinical 129 Modification, 9th Revision (ICD-9CM).”

R:  Thank you for your very instructive comments. We have followed your instructions to revise our manuscript. Please see below.

The medical records associated with live births delivered at Tungs' Taichung MetroHarbor Hospital between January 1, 2013, and December 31,2017, were reviewed, including the paper and electronic records of all infants admitted to the nursery, the sick neonate care unit, and the neonatal intensive care unit (NICU). All diagnoses were determined by 128 qualified pediatricians according to the International Classification of Diseases, Clinical 129 Modification, 9th Revision (ICD-9CM). All diagnoses were determined by qualified pediatricians according to the International Classification of Diseases, Clinical Modification, 9th Revision (ICD-9CM). Infants diagnosed with meconium aspiration (ICD-9-CM code: 770.11);meconium aspiration pneumonia, meconium aspiration pneumonitis, meconium aspiration with respiratory symptoms, syndrome NOS(ICD-9-CM code: 770.11, 770.12, 770.18 ); inhalation meconium of fetus or newborn (ICD-9-CM code: 770.11)with respiratory symptoms(ICD-9-CM code: 770.12);insufflationmeconium(ICD-9-CM code: 770.11 ) with respiratory symptoms(ICD-9-CM code: 770.12), intrauterine, fetal, or newborn aspiration due to clear amniotic fluid (ICD-9-CM code: 770.14);aspiration pneumonitis(ICD-9-CM code: 507); or fetal or intrauterine aspiration (ICD-9-CM code: 770.18)were considered eligible for study inclusion.

Q5: Clinical variables: Authors can refer to it by saying as shown in Table 2 (as this unnecessary reptation)

R: Thank you for your very instructive comments. We have followed your instructions to revise our manuscript. Please see below.

Collected clinical variables were shown in Table 2. Infants delivered prior to 34 weeks of gestational age, with congenital abnormalities, or cyanotic congenital heart disease, or who were immediately transferred to other hospitals postnatally were excluded from the study. Collected maternal data included maternal age, delivery mode, and major medical information [diabetes mellitus, preeclampsia, antepartum hemorrhage, premature rupture of membranes (PROM), polyhydramnios, oligohydramnios]. Collected neonatal data included sex, gestational age, birth weight, APGAR scores at 1 min and 5 min, the need for NICU admission, resuscitation, CPAP use, or ventilator use, and neonatal death.

Q6: Authors need to add reference/s to indicate what guidelines or regulations they followed “All protocols used in the human study were performed in accordance with relevant guidelines and regulations

R: Thank you for your very instructive comments. We have followed your instructions to revise our manuscript. Please see below.

All protocols used in the human study were performed in accordance with relevant the SPIRIT guidelines and regulations the ethical standards as laid down in the 1964 Declaration of Helsinki and its later amendments or comparable ethical standards[54].

2.2. Cell study

2.2.1. Preparation of Meconium

As the birth canal is not a sterile environment[55-59], we collected meconium from 10 full-term, healthy neonates delivered via cesarean section to minimize potential contamination during delivery. Meconium was prepared according to a previously published method[60]. In brief, we obtained first-pass meconium samples within 30min of passage, which were transferred from the diaper into a sterile container. These samples were pooled together and processed in a blender to achieve a uniform consistency. After being homogenized with 0.9% NaCl to a 20% (w/v) final concentration, the meconium was centrifuged at 5,000RPM for 20min at 4°C, the supernatant was filtered through an 8-µm filter (Millipore Co., Bedford, MA), aliquoted into 2-mL sterile plastic bottles, and stored at –80°C until use .For meconium collection, a parent's or guardian's permission and informed consent were required. This study was approved by the institutional review board at Tungs’ Taichung MetroHarbor Hospital, Taiwan, ROC (IRB approval No.: 105047). All protocols used during the meconium collection process were performed in accordance with relevant guidelines and regulations[61].

Q: Again the in Vitro part of the study need to be separated from the clinical/ retrospective review of data of infants with MSAF to get the message (conclusion) of the study quickly to reader.

Thank you for your very instructive comments. We have followed your instructions to revise our manuscript, especially the introduction part. Thank you very much.

Reviewer 2 Report

The authors of the manuscript "Approach to the Connection Between Meconium Consistency and Adverse Neonatal Outcomes: A Retrospective Clinical Review and Prospective In Vitro Study" performed a good and significant study on the importance of meconium type and events for viability of neonates. The manuscript is overall well-writen. My only concern was the fact that the authors include results (or conclusions) at the paragraph ending the Introduction section. This should be removed.

Author Response

Dear reviewer 2

Q: The authors of the manuscript "Approach to the Connection Between Meconium Consistency and Adverse Neonatal Outcomes: A Retrospective Clinical Review and Prospective In Vitro Study" performed a good and significant study on the importance of meconium type and events for viability of neonates. The manuscript is overall well-writen. My only concern was the fact that the authors include results (or conclusions) at the paragraph ending the Introduction section. This should be removed.

R: Thank you for your very instructive comments. The introduction part has been revised in the revised version.

A pilot randomized control trial demonstrated a lack of significant differences in the outcomes of mild, moderate, and severe MAS when comparing cases treated with or without endotracheal suction[52], suggesting that meconium consistency has no effect on MAS prognosis; however, based on our own clinical experience, we suspected a potential connection exists between meconium consistency and MSAF prognosis. To explore this potential connection, we first examined clinical data of neonates born with meconiumfrom a local teaching hospital. 8,316 neonates. Adverse fetal outcomes, including low APGAR scores at 1 min and 5 min, the need forneonatal intensive care unit(NICU) admission, resuscitation, continuous positive airway pressure (CPAP) use, or ventilator use, and neonatal death were primarily idenfied among, suggesting that thick meconium represents a risk factor for neonates who undergo resuscitation. Furthermore, we developed an in vitro model using human alveolar epithelial and bronchial cells to determine the effects of meconium exposure on lung cells. Our findings show that thick meconium is a risk factor for neonates who require resuscitation, and inflammation appears to serve as the primary mechanism for meconium-associated lung injury. human meconium exposure significantly reduced the viability of alveolar epithelial and bronchial cells, and the severity of the response was correlated with both the exposure time and the meconium concentration. In human pulmonarycells, thick meconium induced the production of nitrite, which was negatively correlated with the viability of lung cells. L-Nω-nitro-arginine methylester[L-NAME, a nitric oxide synthases (NOS) inhibitor], dexamethasone, and NS-398 [a specific cyclooxygenase 2 (COX-2) inhibitor] significantly reduced the meconium-induced release of nitrite, suggesting that meconium is a potent inflammatory stimulus that triggers the overproduction of nitrite in lung cells, which may play an important role in the pathogenesis of MAS.
